# "Sun brings all things": Sun and moon lore as biocultural knowledge on Aneityum island, Vanuatu

K. David Harrison[1,2*], Neal Kelso[3], Dominik M. Ramík[4], Nadine Ramík[4], Gregory M. Plunkett[2], Reuben Neriam[5], Wina Nasauman[5], Wopa Nasauman[5], Michael J. Balick[2]

**1** UNESCO Chair in Environmental Leadership, Cultural Heritage, and Biodiversity, VinUniversity, Hanoi, Vietnam, **2** Center for Plants, People, and Culture, New York Botanical Garden, Bronx, New York, United States of America, **3** School of Life Sciences, University of Hawai'i at Manoa, Manoa, Hawaii, United Sates of America, **4** Independent Scholar, Lowanatom, Tanna, Vanuatu, **5** Independent Scholar, Anelcauhat, Aneityum, Vanuatu

* harrison@swarthmore.edu

## Abstract

Across the Pacific, traditional myths and contemporary narratives describe the origins, animacy, and importance to daily human activities of the Sun and Moon. In Vanuatu, Indigenous local knowledge systems interpret ways that the Sun and Moon interact with humans and plants to achieve productive and sustainable lifeways. In this ethnographic study, we explore how residents of Aneityum Island perceive and narrate the Sun and Moon's interactions with animals, humans, and plants. We consider the influence of the Sun and Moon on domains of daily life on Aneityum, including agriculture, architecture, fishing, health care, navigation, time-reckoning, and diverse ritual activities. Aneityum islanders possess generationally accumulated understandings of their relationship to the environment, framed within the local cosmology and communicated orally. Sun and Moon lore—as expressed through myths and stories—directly informs Aneityumese people's actions and efforts at sustainable living, survival technologies, and biodiversity conservation on land and sea. This body of knowledge reveals the causes and manifestations of natural phenomena, and strategies for responding to their impacts. Due to the influences of globalization, many biocultural tools that focus on Sun and Moon lore are at risk of being forgotten. The Aneityumese people—aided by outside experts—are undertaking efforts to document and revitalize this knowledge to ensure the continuity of their resilient and sustainable lifeways.

"Myth is thus an essential form of orientation, a form of thought, and, indeed, a form of life." [1:189]

**Data availability statement:** This manuscript's minimal data set can be found via S1 Appendix. Additionally, all plant specimens may be accessed in digital format at the CV Starr Virtual Herbarium, New York Botanical Garden, https://sweetgum.nybg.org/science/vh/, and in hard copy at the New York Botanical Garden Steere Herbarium.

**Funding:** This material is based upon work supported by the National Science Foundation under Grant No. 1555657 (PI Michael J. Balick) and Grant No. 1555675 (PI K. David Harrison), and by Velux Stiftung Grant No. 1288 (PIs Michael J. Balick, K. David Harrison, and Gregory M. Plunkett). Publication was supported by Swarthmore College. The funders had no role in study design, data collection and analysis, decision to publish, or preparation of the manuscript.

**Competing interests:** The authors have declared that no competing interests exist.

## Introduction

Aneityum island, Vanuatu, is home to ancient sun petroglyphs, a common motif in many Oceanic cultures, linked to creation myths, seasonal cycles, celestial navigation, and beliefs in a powerful sun deity [2–5]. In the 21st century, Aneityumese people continue to reproduce sun motifs in their modern designs, such as woven textile bags, sand art, woodcarvings, and paper drawings. They narrate Sun and Moon myths that were handed down through oral tradition and are still told today, recounting the role of celestial bodies in the mytho-historical past, including during the creation of the world. Myths and knowledge about the Sun (*nagesega*) and Moon (*inmohoc*) are part of *kastom*, a Bislama term denoting the ni-Vanuatu (originally a name of colonial origin, and now widely used as an endonym) "efforts to preserve their heritage and maintain distinctiveness in the face of dominant outside forces" [6:172]; it has also been defined as "festivals, along with any traditional or local practice, style, or belief" [7:5]. *Kastom* includes a diverse repertoire of biocultural tools [8] for survival and provides insights into how the Aneityumese people perceive and steward their island environment [9]. Sun and Moon lore [10], we propose, is more than just a collection of stories and should be recognized as a manifestation of a sophisticated level of environmental intelligence that can enable climate resilience, adaptation, and sustainable living [11].

The earliest human settlement on Aneityum is dated to 2,900 B.P. (950 BCE) [12]. The island was sighted (but not visited) by Captain James Cook in 1774, on his second voyage to the Pacific aboard the HMS Resolution. Recorded visits to the island by Europeans did not begin until 1830. The pre-contact population circa 1830 is estimated to have been nearly 6,000, as evidenced by an extensive irrigation and agricultural terrace structures, although some estimates range higher [12]. The island was divided politically into seven chiefdoms, which engaged in occasional warfare or competitive feasting and food exchanges. The population declined to below 200 in the early 20th century due to the effects of introduced diseases and blackbirding, but is now around 1,000, mostly inhabiting the coastal perimeter.

The first Christian missionary work in Melanesia occurred in southern Vanuatu, in the province now known as Tafea [13], which includes the island of Aneityum. From the 1840s, European and Polynesian Christians established permanent missions on the island [14]. These missions saw varying degrees of success; Aneityum's population became predominantly Christian in a shorter time, and with relatively less conflict, than the population on Tanna, an island just to its north [15]. The fact that Tanna had a larger and more linguistically diverse population at contact almost certainly played a role in this difference. However, Douglas [15] gives more weight to the fact that the cosmologies of Tanna and Aneityum sit at two ends of a spectrum that varies across Melanesia. Tanna at the time of European arrival had a cosmology in which the spirits were seen predominantly as subservient to humanity. Effective magic and ritual depended more on the ability of the practitioner—the spirits were expected in most cases to bring about what a skilled practitioner requested. On Aneityum, however, "spirits [known as *natmas* in Anejom̃] were accorded some freedom of action in human affairs, played a regulatory role in relation to human morality and

were approached through prayer, sacrifices and bargaining" [15:16]. The many *natmas*—ancestral spirits and the spirits of plants, animals, stones, and celestial bodies—were the actors that ultimately brought about most events that affected the world.

Given this background, Aneityum's pre-contact religious and cosmological beliefs may have been seen as more compatible with the missionaries' message. Gardner [16] finds in early missionary texts that the Christian missionaries, though successful, were somewhat perplexed by the reasoning behind the changes in local beliefs. She contends that the process was not one of pure linguistic or cultural relativism, in which Aneityumese people only understood the missionaries' message within their own pre-existing frameworks, but rather a two-way process of translation. The Aneityumese came up with a new cosmology in which old and new religious beliefs were syncretically combined.

In other analyses, the conflict inherent in missionization is more apparent. Mitchell [17] argues that missionaries intent on converting the Aneityumese population did so in part by changing their conception of and organization of time. Pre-contact time was the native, cyclical time in which the events of the mythic past were constantly recreated through rituals, feasting, and preparation for feasting. In this analysis, it is important to note that the only written records of the process of conversion were in each case written by the missionaries themselves. They very likely do not paint a comprehensive picture of these cultures at that time. Contrary to the missionaries' teachings, the Aneityumese people took actions to preserve certain aspects of their *kastom* in secret. Our colleagues on Aneityum talk of underground *nakamals* (ceremonial meeting places), concerted efforts to preserve the old *kastom* in remote areas, and the decision to move many of the sacred stones to the nearly unpopulated eastern side of the island. To this day, the eastern half of Aneityum is the least densely populated by people, but rich in sacred stones and other culturally important sites.

Other researchers have reported similar conflicts between missionary accounts and local actions. Though some local people reportedly partook in the destruction of sacred stones and groves during the early decades of missionization, they also passed down oral histories of the locations and significance of these stones. In a remarkable study, Bedford et al. [13] chose an archaeological dig site based on local oral history of the desecration of Aneityumese "idols." For over 150 years, local people had passed down the exact spot of ground under which sacred stones were buried, and they are now once again above ground. In reality, many other aspects of Aneityum's pre-Christian culture were passed down to the present. Local people, including three co-authors of this paper, continue to engage in concerted efforts to revitalize and strengthen these practices, including the traditional chiefly governance system [18].

In part due to local preservation efforts during and after missionary times, many sacred stones are still standing Fig 1. A number of these are carved with images of animals and sun motifs drawn in several different ways Fig 2. Concentric or plain circles with sun-like rays extending—although absent from the inventory of common signs found in the cave art of Ice Age Europe [19]—are a frequently attested motif in the western United States [20], Polynesia [21,22] and Melanesia [23]. In the Aneityum petroglyphs, the sun disk may be a spiral or a set of concentric rings, with or without emanating solar rays [24]. Prehistoric rock art showing anthropomorphic standing sun figures with legs (and sometimes arms) are attested in Canada [25] and Kyrgyzstan [26]. Something apparently unique to Aneityum is that the Sun is portrayed with two legs of uneven lengths Fig 3. Spriggs and Mumford [24] report that on Aneityum the sun and moon are identified with a pair of sacred rocks (of which only the "sun" rock bears petroglyphs (Fig 4), while the "moon" rock is not inscribed). Human contact with these rocks is said to be efficacious in fishing magic (for men), and mat-making skill (for women).

The image of the Sun with legs—named *Nagesega Atga*, the Walking Sun, in the Anejom̃ language—is thus of central importance in the cultural chronology of Aneityum, and to our analysis. Our Aneityumese colleagues describe it as a remnant of a past Sun-worshipping culture on the island and say that it symbolizes both the traditional governance system and the traditional calendar. With the Walking Sun as a keystone cultural symbol, we discuss the importance of the Sun and Moon in the biocultural landscape of Aneityum. Through oral myth and history as well as descriptions of present practices, we demonstrate that the Sun and Moon remain important figures in Aneityum's cosmology, which guide people's interactions with the land. Though the process of Christianization has significantly affected the cosmological landscape of

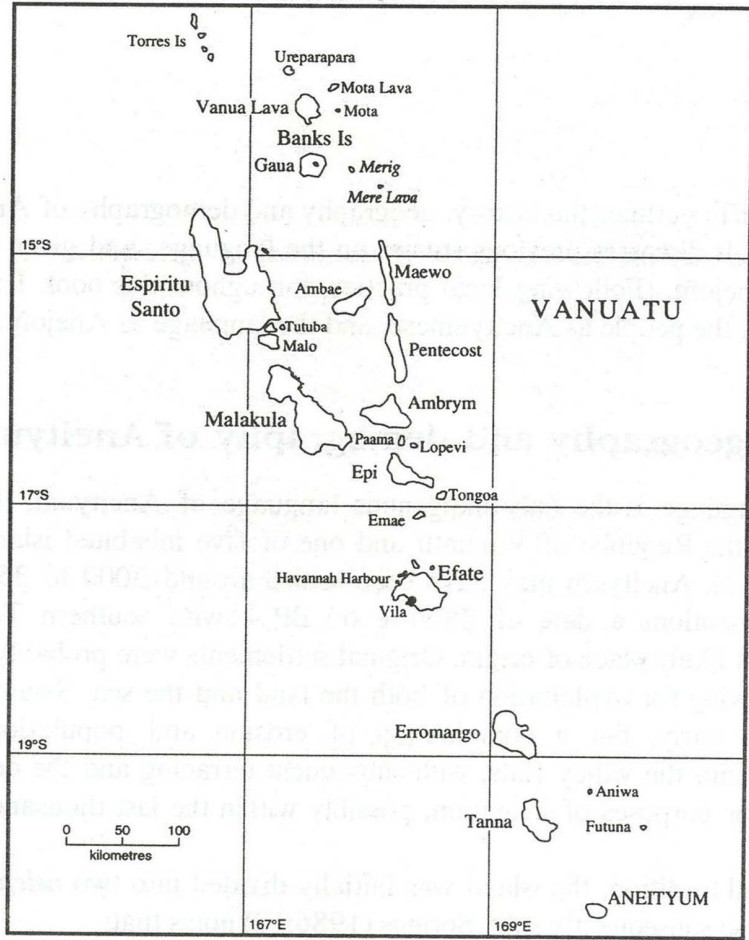

**Fig 1. Map of Vanuatu showing the five islands of Erromango, Aniwa, Futuna, Tanna, and Aneityum.** Reprinted from https://commons.wikimedia.org/wiki/File:Map_of_Vanuatu.jpg under a CC BY-SA 4.0 license, with permission from Youraveragejohndoe, original copyright 2016.

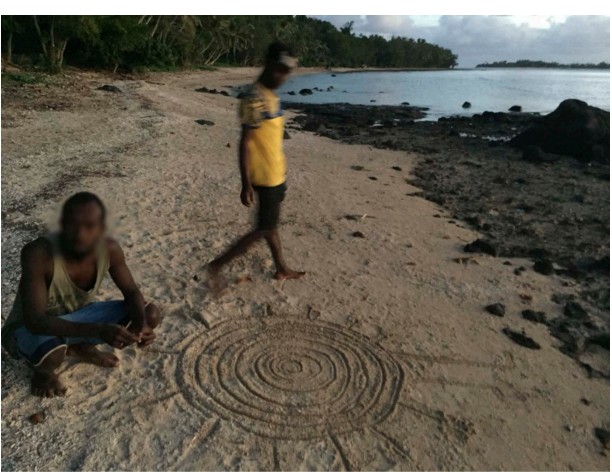

**Fig 2. A sand drawing of *Nagesega Atga* with seven concentric circles and twelve rays created by Orien Namu (left) and Chris Nevehev (right) at Anelcauhat (photo K. David Harrison).**

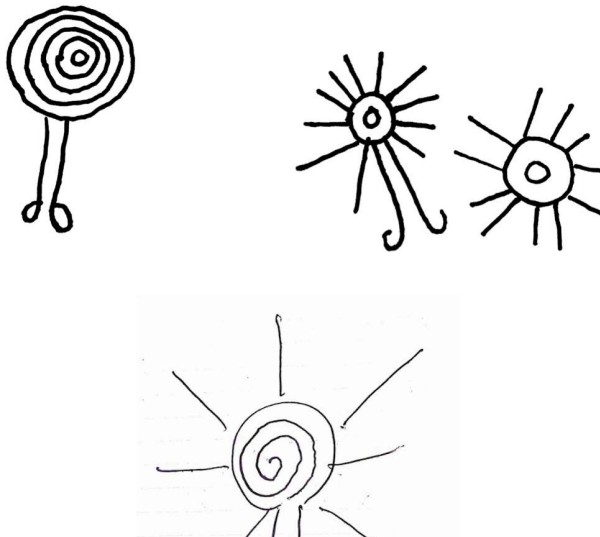

**Fig 3. (Upper) three sun petroglyphs as documented by Spriggs and Mumford [24], redrawn here as similar but not identical to the original images and for illustrative purposes only, and (lower)** *Nagesega Atga*, **the Walking Sun motif, drawn from the authors' 2017 field notes.** Each of the walking sun images has differently shaped feet. (Drawings by the authors).

Aneityum and the way that local people conceptualize time and the world, at present this landscape is still characterized by uniquely Aneiyumese beliefs and practices.

## Methods

We conducted ethnobotanical and ethnographic interviews on Aneityum in 2016, 2018, 2019, 2020, and 2024 as part of the *Plants and People of Vanuatu* project. Working closely with local cultural experts (three of whom are co-authors on this paper), we accessed and recorded local knowledge narrated in the Anejoṁ [aty], Bislama [bis], and English languages, as all three are used routinely by most islanders. While conducting transdisciplinary fieldwork and investigating Indigenous beliefs about the Sun and Moon, we developed several research questions based on our initial conversations with local experts: 1) How does the Sun/Moon influence the activities and livelihoods of people on Aneityum Island? 2) What agency is ascribed to Sun/Moon in local tradition? 3) What do stories of creation and other mythical events reveal about how people perceive the Sun/Moon in Aneityumese culture? 4) What domains of knowledge (e.g., architecture, botany, wayfinding) are influenced by beliefs concerning Sun/Moon?

Following a detailed explanation of the project to the community and their traditional leaders, knowledgeable individuals (whom we refer to as local cultural experts) were identified through network sampling [27]. We discussed the project in greater detail with the local cultural experts who agreed to participate and obtained prior informed consent from them as a condition of their participation. We recorded these interviews in digital audio-video and took photographs and handwritten notes for later analysis. Our primary method is the collection and co-curation of knowledge that remains the intellectual property of the Aneityum community, rather than data extraction. As such, we include local Indigenous experts as co-authors in our published works. The scope of our research on Aneityum was environmental knowledge broadly construed, transmitted orally or via cultural practices such as architecture, canoe-building, gardening, navigation, plant-based

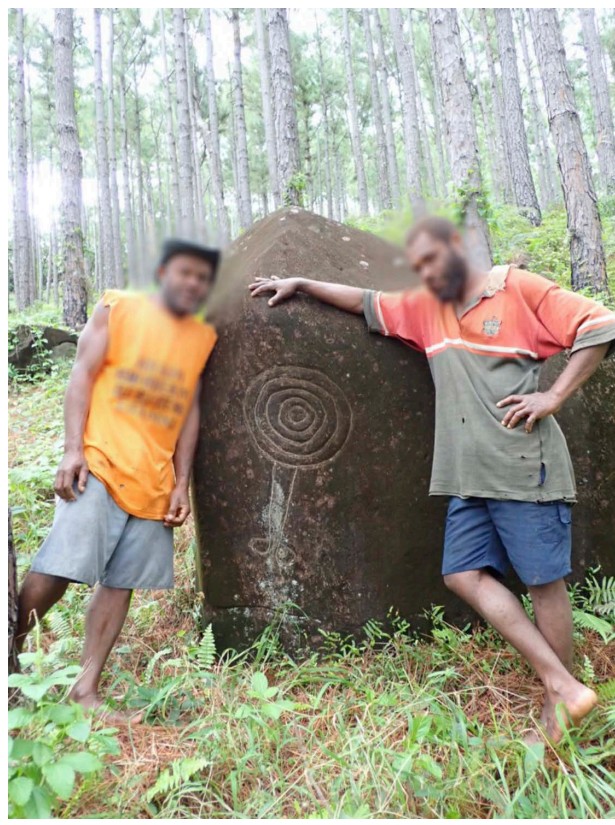

**Fig 4. Aneityum islanders Wopa Nasauman (left) and Kelly Makoy (right) with a sun petroglyph found near the village of Umej, featuring five concentric circles and two legs.** (Photo Gregory M. Plunkett).

medicine, wayfinding, and weather magic. A significant part of our project, as reflected in the list of local plants and plant uses in S1 Appendix, was collecting and identifying plant specimens, which we later deposited at herbaria in Vanuatu, the United States, and elsewhere [28]. To elicit uses of plants, we showed collected specimens (and/or photos) to our local consultants. We also recorded the lexicon and created an Anejom̃-English Talking Dictionary [29]. The Swarthmore College Institutional Review Board granted us an exemption for this work. Vanuatu governmental authorities in Port Vila granted us a research permit. Key outputs include a large corpus of recorded narratives and myths, which we sample in this paper.

## Results

The key to understanding the Aneityumese people's cosmology and relation to nature lies in their cultural narratives. These narratives reveal a mythico-historical past [1], but also influence their lives in the present with respect to human relations to the environment:

> At the beginning of time, the spirits of darkness and light were fighting to be in charge. A spirit called *Nagesega Rada*, the Sun God, made a decision to give one part of the day to the Sun, and the other part of the day to the Moon. (Told by Reuben Neriam, of the Southern Paramount Chiefdom.)

> Sun and Moon were boasting to each other about who is greater. Moon said he was more powerful indeed. But Sun said he was shining more. They got into a dispute. *Inwou an Nadiat* [the Milky Way] said that he would resolve their

dispute by separating them. Sun was to shine during the day and Moon during the night. Thus, the problem was solved. (Told by Wopa Nasauman and Thomas Japanesi Lalep of the Northern Paramount Chiefdom.)

In the Anejom̃ language [see 14,29,30], the division between night and day at sunrise is called *nacsei* (to cut or to separate). Each day, at about 3:00 am, when there is a small amount of light in the sky, *Nagesega Rada* makes *nacsei* again, to tell the night that its time is almost over and that it is almost time for daylight to come. The sunset is referred to as *nacai melmat* (green spear), the separation at the end of the daylight. Separations of night and day, sunrise and sunset, north and south define not only temporal and geographical relationships on Aneityum, but also social ones.

Aneityum's traditional governance system is one of the key ways that islanders identify themselves in relation to other islands, and within the geographical and familial relationships on the island. In this system, the entire island is now divided into two Paramount Chiefdoms, the South—which is geographically the eastern half of the island—and the North—the geographical western half. Wood [31:105] records that "the [North] is known as the "Sunset Moiety" (*Nelcau-sokou* or *Nelcau-Inpekeritinpeke*), and the [South] was conversely the "Sunrise Moiety" (*Nelcau-jekou* or *Nelcau-Anejom*)". These Paramount Chiefdoms are sub-divided into six or seven subordinate chiefdoms, which are then further subdivided into districts. The divisions at each level are called *nelcau*, a word that has as its primary meaning 'canoe, boat, ship', and by metaphorical extension, also 'district, kingdom' [14:99]. Vital to this system are the Holy Priests, individuals who maintain and pass on magical and ritual knowledge. These are called *Ilp̃atimarid Itap* in the South, or *Ilp̃usohos Itae* in the North.

In the past, practitioners within this system were not necessarily hereditary, but rather were chosen through the process of "nomination," or name bestowal [31]. In this system, chiefly positions of power were typically conferred from fathers to their sons or other male relatives, though this was not a strict rule. Colonization and the formation of Vanuatu as a unified nation have eroded this practice to a degree, but it continues to this day. Totemic (district) identities are still maintained through nomination, though they are mostly social when once they were closely tied to land.

The governance system is represented graphically in drawings of *Nagesega Atga*, the Walking Sun. In an account from Umej village, the center of the Walking Sun symbol is the chief, and the outer circles are his sub-chiefs. Other narratives we recorded state that the legs of the Walking Sun represent the two Paramount Chiefdoms, while the solar rays emanating from the disk represent the six or seven chiefdoms. The legs are of unequal length and, according to Reuben Neriam of the South, they represent two spirits that look after each half of the island. The longer leg is *Nagesega Rada*, a tall spirit whose name means "we worship the spirit of the Sun God." *Nagesega Rada* is associated with the Southern Paramount Chiefdom and comes from *Um̃ai Nupni* (Matthew Island). The shorter leg is *Nowaneañ Anivat*, a short spirit and the originator of the coconut. *Nowaneañ Anivat* is associated with the Northern Paramount Chiefdom and comes from *Um̃ai Neañ* (Hunter Island). Both Hunter and Matthew are small islands about 330 km to the SE of Aneityum; they are uninhabited and claimed as part of Vanuatu's Tafea province. The powers of the Holy Priests are said to come from these respective islands, with which Aneityumese people have had a relationship for many generations.

Following is a collection of seven traditional stories from Aneityum recently recorded by our research team that involve the Sun and/or Moon as world-building entities. The stories touch on the origins of the people of Aneityum, the introduction of Sun worship, and the origin of petroglyphs, and fire. When relevant, we note the Paramount Chiefdom (Southern or Northern) from which the details reported here originated. There is still a strong social division between these units, though it is eroding. Some beliefs and practices may be held by just one or the other Paramount Chiefdom and may not be shared freely between individuals from different sides.

### On the origins of Aneityum's present population

This narrative was performed jointly by Wopa Nasauman of the Northern Paramount Chiefdom, and Reuben Neriam of the South.

The first people of Aneityum were called *Nupua Naiwanma*. They were dark-skinned and strong people, but very short, as evidenced by the lengths of their gravesites, which can still be seen in places such as Anecro. The presence of graves suggests that they practiced burial of the dead, unlike later people. They lived only in the bush and never went to the saltwater.

The second wave of people were called *Matua Ailed*. *Matua* is a Polynesian word of unknown meaning, and *ailed* means "paddle." These "paddle people" arrived in Aneityum on canoes. They were tall but had "yellow" (pale) skin. They brought with them the taro irrigation system and stone walls. After arriving in Aneityum, some of these paddle people moved on to northern and southern New Caledonia. This second wave of people were Sun worshippers. The name Keamu for Aneityum is thought locally to be of Polynesian origin, whereas the word *nijom*, which gives the current name for the island, means "flower of the coconut." After the *Matua Ailed* arrived, the *Nupua Naiwanma* moved to the western side of the island.

Although the meaning of *matua* was unknown to our Anejom̃ speaking consultants, the word in Polynesian languages can signify: 'mature', 'parent', 'elder', 'old man', or 'main' [32]. A cognate term *matak* (meaning 'cooked' or 'ripe') is attested in Netwar and other Tannese languages [33]. Thus *Matua Ailed*, the name of the second wave settlers, could be translated as 'the paddle elders'.

The narrative continues, explaining that a later wave of Polynesians arrived (also "paddle people") and settled in the central third of the island, forcing the first wave farther west and the second wave to the east. They had the same culture as the second wave (building irrigated taro fields and stone walls) and were the last people to paddle to Aneityum. These people provide some connections to the people of nearby Futuna Island, including the story of Riu and Warariki. As to what happened next, there is a divide in opinion that remains to this day between the people of the "South" (the eastern half of Aneityum) and those of the "North" (the western half). In the South, people believe that the Nupua Naiwanma and the Matua Ailed people mixed, producing the current population, who are dark-skinned and strong (like the *Nupua Naiwanma*), but tall (like the *Matua Ailed*). People from the North believe either that there was no mixing, and/or that they refused to mix. Due to this difference, they separated themselves from the mixed-race people [or the people who claimed there was mixing] and settled in the North. This divide is maintained in the traditional governance system and also resulted in people from the two halves of the island maintaining separate ways. As an example, Reuben Neriam stated that his father (Southern) and Wopa Nasauman's father (Northern) generally did not talk to each other. It is only today that the separation is being overcome, and people like Reuben and Wopa now interact very regularly. Prior to the missionaries, Aneityum people after the Second Wave believed they originated from the sea, and therefore, when they died, their bodies were not buried in the ground but cast into the sea. After death, their spirits would either go to a good place (if they were good people) or be tormented and beaten by the stones (if they were bad people). A legend (not included here) that we recorded in Umej describes the complex path of the dead and the trials they must undergo before reaching a final resting place.

## A northern story of the split into two paramount chiefdoms

Wopa Nasauman and Thomas Japanesei, both of the North, provide a more detailed story of why and how the people of the North and South came to be separate:

The Moon (*inmohoc*) is associated with women, who are associated with the *Nupua Naiwanma* (the first people of Aneityum), whereas the later "paddle people" (*Matua Ailed*) are associated with the Sun (*Nagesega*), which represents man. The Moon people were the first to arrive, coming from someplace in the South, and brought stones, including stones for wind directions. These were small, dark-skinned people. The Sun people arrived later, by the westerly winds, and they brought another set of stones, especially for controlling the Sun and weather. They were tall and light skinned. These two peoples did not know each other.

At one time, the Moon people hung a leaf [used in weather magic] from a particular type of tree in order to keep the wind coming from the same direction. Then a *nipan sensen* (canoe with sail made from woven *Pandanus*) landed at Anijiau. At this point, there are two caves, one called *Inmaipan*, which was male, and the other *Inmanavanua*, which was female. These are associated with two hills, *Kiji* (female) and *Nasuanelco* (male). *Kiji* is connected to the *kastom* road that goes to *Inrero Atahañ*, while *Nasuanelco* goes to *Inrero Atamañ* (these are the female and male peaks of Aneityum's highest mountain, Mount Inrero, respectively). The Moon people cross Mount Inrero and end up along the northern coast of Aneityum at the area called Anace; this is the place where the Sun people were living.

While at Anace, the Moon people went to a river to swim, and there the Sun people were also swimming. They asked each other if they were the same or different, and they decided that they were the same, but that they should live separately. Afterwards, the Moon people returned to the Anelcauhat area, and were thereafter associated with wet places, while the Sun people in the North were associated with the dry places. At one point, the two peoples began to intermarry. If people from the South want to make dry weather, they will bring a leaf from the North and make *kastom* (and vice versa).

Moon people made rock carvings of birds, snakes, turtles, fish, etc.—especially things coming from the sea. By contrast, the Sun people carved the Sun, which is sometimes just a plain circle (as at Anumeij in the North), or sometimes the Walking Sun (either with the interior spiral, or with the exterior rays, as at Umej).

The Moon people did not worship the Sun but only did gardening. The Sun people worshipped the Sun at the sites where the carvings are located.

## A second-hand account of Aneityumese sun worship

This version of the story was told by the late David Nasauman, of the North, who was retelling his grandfather's account of Sun worship. In his account, Sun worship was not confined to one Paramount Chiefdom, but was rather divided based on when it was performed: sunrise or sunset:

> In past times, as witnessed by my grand-father, people from around Umej (those in the Southern Paramount Chiefdom) would wake up before the sunrise and go to Umej to worship the Sun when it rises. They would thank it for their life, for their families and their children, for the new day that is coming, for the food that is already there and that will come, for their gardens and the fishing—they attribute all of this to the Sun as their god.
>
> Those living in the area of Anawense (those in the Northern Paramount Chiefdom) would perform the same worship but at sunset. They would ask the Sun to preserve their lives until the next day and ask that it comes with even more things than it brought on the previous day.
>
> Both in Umej and Anawense this would be performed daily by all people. It was believed that the Sun was the one who brings all things to the people, the food, life and so on. This was a very strong belief; people were very devout in praying to the Sun for what they needed. When the missionaries came, they made sure to establish their presence in both Umej and Anawense as those were the centers of the traditional beliefs.

## On the hierarchical position of Nagesega Rada, the Sun God

This version of the story was recounted by Reuben Neriam, of the Southern Paramount Chiefdom. Neriam makes no claim as to who participated in Sun worship but does imply a hierarchy to the gods and spirits of Aneityum's pre-Christian cosmology. To him, the Sun God, *Nagesega Atga*, is the "main god":

Where I come from, our ancestors—but we no longer use it now—were sort of Sun god worshippers. They came out to a place where there's a big [area] without trees growing in it, and then before the sun comes up you get there first. And then before the beams come up, that's when you practice the worship. You find the knees [kneel down], put the knees in the ground, you stay there maybe 1–2 hours, and then get back to where you come from.

And they believed that the god—the Sun god—is *the* god, [for] which we have a name: *Nagesega Rada*. *Nagesega* is Sun, and *Rada* is something like 'god worship.' He will answer your belief, or he will give it to you… I believe that the Sun is also a god, power of God. Where I come from, we have Sun gods, we have thunder gods, we have rain, but they are under the Sun god. We have the gods, everything—they have their powers… but the main god is *Nagesega Rada*."

This is only an excerpt of Reuben's story, in which he reconciles these pre-Christian beliefs with his contemporary understanding of Christianity. For him, there is no inherent conflict in these cosmologies.

These stories establish that the Sun is an incredibly important figure in the past and present cosmology of Aneityum. Whether the island truly had a Sun-worshipping culture reminiscent of transcendental religions is addressed in the discussion below. The rest of the results include traditional stories in which the Sun and Moon are actors. We then discuss contemporary practices that involve the Sun and Moon.

### The story of Kagai

In the following story, told by Wopa Nasauman and Thomas Japanesei of the Northern Paramount Chiefdom, the Sun and Moon conspire to kidnap an old man:

In the east of Aneityum island, in a place called Idei, there was an old man called Kagai. He went to the sea and climbed on the coral rock and jumped into the water to grab some lobsters. But he was surprised that it was not lobsters [that he found there] but the Moon, which then carried him away. The Moon carried him to Um̃ai Neañ and Um̃ai Nupni, that is Matthew and Hunter Islands—it is another world.

The Sun hadn't come up yet. Then the Sun came up and the Moon asked him: 'Guess what I found.'

The Sun replied: 'I don't know, but if it is a man, you ought to tell me.'

But The Moon hid the old man in her grass skirt. For one month the family of the old man was searching for him but couldn't find him. And after one month, [they] considered him lost. The old man was living with the Sun and Moon and ate only bananas and sugar cane. The *makas* [spit-out fibers] of chewed sugar cane formed a line from Matthew and Hunter Islands back to Aneityum. But at one point, the old man was brought back. His stone is still there on the reef now, and people were surprised to see him. He went to his village, but the Sun and Moon told him not to say that the Moon took him away. But after some time, the old man forgot and told his family that it was the Moon who took him away. Then his family told him to join them to collect some coconut leaves to block a tidal pool and go fishing. So, they made a net from coconut leaves and the old man was on the end of the net. The Sun went down as a fish and cut the man, killing him because he gave away the secret. The stone of the man is still there.

### The yellow men of Ihili

This story, told by Freddy Tamana of the Southern Paramount Chiefdom, tells of the origin of Aneityum's solar petroglyphs. A version of this story is also recorded in Spriggs and Mumford [24]. According to the story, the solar petroglyphs were carved by *Matua Ailed*, a wave of people who came to the island and are now believed locally to be Polynesians:

In this place [Ihili], there were some men that came ashore here… They came ashore here, but when the elders saw them, they were yellow. They were here for a while, but at night they made fires and did *kastom* dances around the

fires. They did *kastom* dances around the fires. During the day, they all went on top of the hill over there [towards Ihili] and were carving stones. If we have time, we can go look at the marks they left carving the stones. They made Sun drawings on them. One of its legs is long, with… the same design, all of them, and drawing the mark of the foot on it. That was their work now during the day, they went on top to make this work. Carved it…there are lots of designs up over there.

In the day, they carved them. In the night, they came down to do this work: They made fires, they did *kastom* dances, they went around the fire. Then, the elders that were here before said, 'These men are dancing and spoiling our kava here. It would be good if we killed them.' Well, the chief made this speech that they would go kill them. Then they went and they killed them. Just like this. This is a story that the elders passed down and eventually passed to me. But we will go look at their marks, that they carved on the stones here.

**On the origin of fire**

In another story told by Tamana, he explains how fire came to humanity from the Sun and Moon:

This is the story of Sun and Moon. A long time ago, the elders say that they didn't have fire. They didn't have fire to bake food. Every time the chief gave a speech, the men would carry taro, kava, they carried it to him. They carried it to the *nakamal*. They carried it to the *nakamal* to eat. But when they ate, the taro was very sour… They didn't cook it. They ate it just like this. They dug it and brought it out from the garden, and they ate it…

But when Moon carried his food and came with Sun, their food was just fine. It wasn't sour. Now the chief and the men started thinking. They said, 'Now why isn't the food of those two sour? Their taro isn't sour?' Immediately, they set a plan. There were two men that went to check, saying 'What is it now that is making the food of those two not sour like that?'

Well, they sent one man named Lecap Lecap, another one named Jumas Juma. The two went. The two went by flying. The two flew to the home of Sun and Moon. Well, the two had a place but because it was a long way away you and I couldn't reach it… in the East.

Well, the two flew to the house of those two, but their house—Moon was inside making a fire. He was making a fire, because for a long time there wasn't fire. The men were making taro, but it was sour. They were eating food, but it was food that was not ripe. So, Jumas Juma looked with Lecap Lecap. The two of them looked inside but they saw smoke from the fire coming out from their house now. From a hole in the stone.

So, the two made a plan. Jumas Juma made it. He said, 'The two of us will make wind go inside.' So, they made wind, they blew into it, and it went inside. It went to Moon because Moon was making his *laplap*, he was baking his taro. Well, when his eyes [looked]—just like this—because the fire went inside and was sour!

He got up, he thought he would carry wood and put it into the hole in the stone, but he threw it outside. After, Lecap Lecap flew down and picked up the stick fire [a kind of wood that burns slowly with red embers used to light the path when people return home at night after drinking kava, as well as for other purposes including transferring fire from one location to another, lighting tobacco, etc.], then went up. When it got hot, he threw it. Jumas Juma picked it up again. He flew up, and when it was hot, he threw it. The two of them did this work there, carrying it up. When it was too hot, they lost it. The fire fell down and burned the place [a physical place atop the cliffs on the southern coast]. Looking now, you will see the spot here. Trees can't grow there. They say that they burned it, they burned the place here and people on Aneityum looked and only saw fire here. The men just looked and said, 'Ah, fire, this is something that Moon and Sun, those two, used so that their food was done, and they ate.'

They made it so that when you look at this spot today, wood [trees] can't grow there. The place of Sun and Moon is here. So, this is the small story of Sun and Moon, that they say those two are in this place.

## Contemporary practices

The *taɱava* ritual—the ritualized spitting of kava—is a central part of life on Aneityum, performed each day when men gather to prepare kava and socialize (cf. Lindstrom [7] writing about Tanna Island). It is illustrative of the perceived cosmological relationship between humans, the Sun, and the Moon. The following description of *taɱava* was given jointly by Wopa Nasauman and Reuben Neriam:

> When the Sun rises, man must work. Before the Sun sets, all the men from a certain "tribe" or "clan" (e.g., the tribe for taro, or for yam, or for kava) gather together in the *nakamal* to prepare kava and tell stories. These represent "reports" about what is happening in their gardens. Before sunset, the chief of the tribe [or] clan, called the *Natimarid Uilp̄aca*, seals the report by making the *taɱava*.

> The word for the ritual is derived from "*tamu eva ren akaija*"—'whatever we seal, that is what we stand for'—which indicates a solidarity among all the men in the nakamal. This must be done before sunset, to allow the Sun to receive the report. When the Sun sets, it carries the report as cargo on its back, then delivers the report to the Moon. The moon then crosses the sky and talks to the constellation of the Pleiades, which is also a spirit called *Ilpu-Halu Comñomoi,* the children of Comñomoi.

> Before daylight returns, the constellation relays the report back to *nepothan*, the ground. If the report suggests things are not going well, then the ground will adjust things to increase the sunlight, or rain, or wind. The ground is the mother, on which all the people depend.

As this narrative illustrates, the Sun, Moon, and ground are linked with humans and other beings in the environment through a rich repertoire of bio-cultural tools used on Aneityum. The following section describes current cultural practices that engage the landscape, starting with the geography of the Sun and Moon on the ground, then describing how the Sun and Moon relate to practices such as timekeeping, architecture, and agriculture. In the scope of our (ethno-)botanical field study, we collected and identified numerous plant species (S1 Appendix) that have traditional uses in relation to the Sun, including: (a) plants dried in the sun to create usable fibers, (b) plant-derived medicines used in relation to the sun, (c) plants used to provide shade for people, houses, or other plants, (d) plants used in foodways, either for consumption or for the preservation of foods, and (e) plants used in weather-magic rituals. Beyond the inventory of useful plants in S1 Appendix, there are dozens of other plants used in architecture, basketry, and material arts that must be harvested at specific times during the lunar/ecological calendar to ensure their quality.

## Geography of the Sun and Moon on Aneityum

Aneityum's sacred geography is infused with the power of the Sun and Moon. They are not only celestial bodies that are removed from humanity up in the sky, nor are they beings that were only present in the mythical past. They continue to reside in the physical landscape of Aneityum, and humans may interact with them through rituals and sacred stones.

The Sun and Moon are, respectively, a man and woman who form a couple. According to Freddy Tamana of Umej, they live in the eastern part of the island in a place called *Itunurau* (lit., 'boiling air', the movement of hot air caused by the sun). Their house is a cave called *Inmanij* (lit., 'black stone'). The Moon's work is to weave mats, and her sacred stones are found close to *Itunurau*. Young girls come to touch the stones to receive the Moon's blessing and be able to weave nice mats.

As mentioned previously, many sacred stones were carried to the eastern half of Aneityum during missionary times to protect them during the widespread destruction of what were considered "heathen idols" by the missionaries. Today, the

eastern half of the island is dense with sacred stones of many kinds, including analogues of Matthew and Hunter Islands, which are of key importance to the tribe of Holy Priests (a hereditary lineage in Aneityum tasked with performing religious rituals). Many of the sacred stones and the particular ways they can be used for magic remain secrets of the inheritors of such knowledge, most notably the present heirs of the Holy Priests. Among the sacred sites found in the east is a hill called *Nedun Necdaduin Cap*, at which the *Namlañhas* ritual to burn the entire island may be performed. Rather than an act of pure destruction, this weather magic ritual [34] falls within southern-Vanuatu conceptions of ecology, in which renewal comes from sporadic destructive events.

The sacred stones that are carved with images of the Walking Sun are not only important in terms of traditional governance but also convey small details of local time-reckoning practices. In the part of the year when the days are short, the Walking Sun walks with its shorter leg. In the long days it walks with its longer leg. In the time of year with short days, the people of Anawense—who live on the western shore and are part of the Sunset Moiety—had to be careful to finish all the chores early to be in time to pray to the setting Sun. In the time of year with long days, life was more at ease because people would know there was more time before they had to go to worship the setting Sun. Starting from the broad seasons of long and short days represented by the Walking Sun, we can look at the complexity of time-reckoning on Aneityum.

### Aneityumese time-reckoning with the Sun and Moon

Local time-keeping strategies are attuned to specific locations on the island. For example, the geography surrounding Umej keeps people apprised of the gradual progression of the Sun throughout the year. When the Sun rises above the hills where *nemek* (turmeric, *Curcuma longa* L., an introduced species) is planted, it is the season of short days and the *nemek* is ready for harvest. When the Sun rises over a place called *Anemek* (or *Nemek Kari*), it is the shortest day of the year, and when it rises over Isia, it is the longest day of the year. On both equinoxes, the Sun rises over Imtaña. As with Tanna's "living Stonehenge", in which people reckon the solstices in relation to the setting sun in relation to the placement of certain trees (for more details, see Balick et al. [35]), this geographical astronomy may be quite long-standing without leaving an archaeological record.

The phases of the moon are named, and each have their own significance. Freddy Tama shared the following names. *Netpoupou* is the new moon (*net* means taboo, *poupou* means down). The first man to see the new moon makes a staccato noise by yelling while hitting his open mouth with his hand to avoid any negative effects. This is also the time when, during the day, it is good to plant crops. *Inmohou ahajen* is the first quarter (lit., part of the moon). When one watches the growing first quarter, it can "cut him behind his ears." *Inmohou owasjei* is the full moon (lit., the moon is plentiful). The time of the full moon is good for planting bananas, which should be done late evening or with the coming night. On new moon nights and on full moon nights, when the moon is not out yet but you already see its light, it is unwise to wander long distances during the night because spirits are particularly active.

Compare these names to those given by David Nasauman and Reuben Neriam. *Nupyehet* is the new moon. This is the best time to burn a living tree because there is no water inside. Living trees were burnt to clear a field (before the advent of knives and chainsaws) by putting firewood and dry leaves at the base. *Tilaconai* is the first quarter moon, while the last quarter of the moon is *Epigjai iran* (lit., darkness added) and *Tilaconai ehteleceinai* is the full moon. Crescent moons on either end of the month are called *Erinmerei* (lit., leaf of *Acacia spirorbis*, which is crescent-shaped). They are said to be portents of rain.

Much of the time-reckoning tradition of Aneityum involves ecological calendars—systems in which people follow plant and animal phenology along with other natural rhythms to keep track of time [35]. Often, individual natural events correspond to specific human activities. For example, when lightning bugs of the genus *Atyphella* (*nepelvanu*) are plentiful, it is time to plant yams. *Nispeheñ* trees (representing an undescribed species of *Dracaena*) provide two signals: when insects start eating holes in the trunk, it is time to clean up the garden, and when new growth comes up after a tree has fallen, it is a good time to plant kava. When *namaka* (*Urena lobata* L.) flowers Fig 5, this indicates that cyclone season has passed.

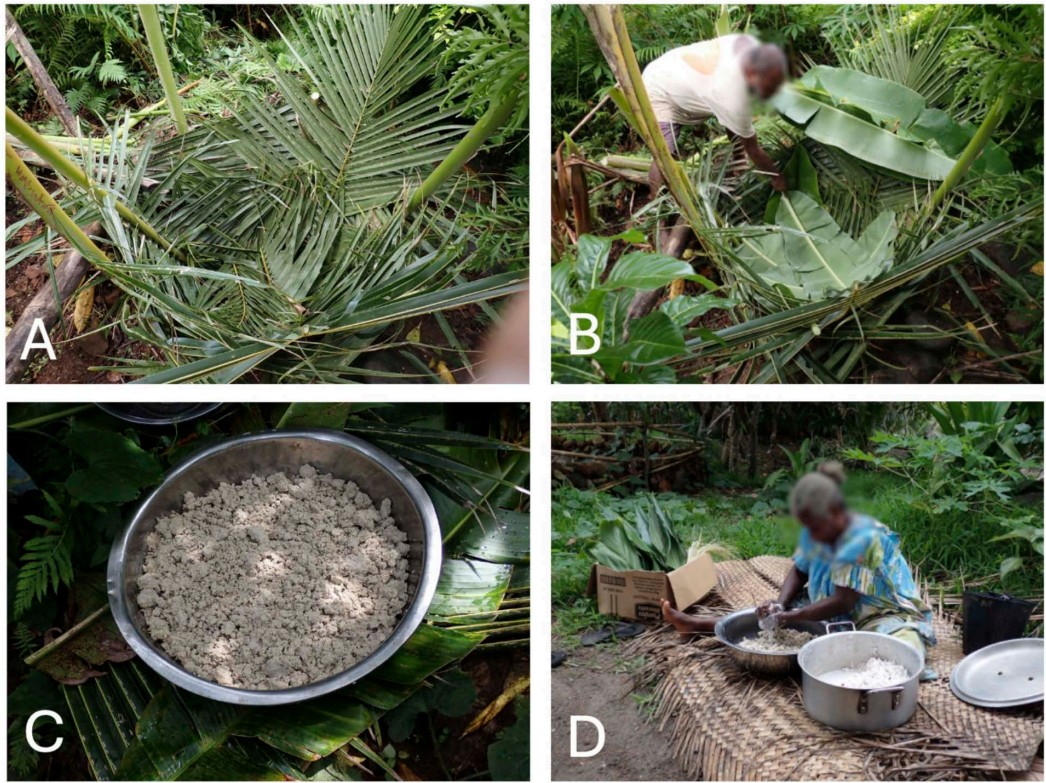

**Fig 5. Namarai.** A: The outer, support layer of the *namarai* pit made of coconut leaves. B: Adding the *Heliconia indica* leaves to seal the pit. C: The fermented product. D: Mixing the fermented product with coconut milk to prepare for cooking. Photos by Dominik M. Ramík.

Alongside these ecological calendar plants is the lunar calendar based on *inmohoc*, the moon. The Anejom̃ lunar calendar is somewhat unique in the islands of Melanesia in that it is based predominantly on grasses, including wild cane (*Miscanthus floridulus* (Labill.) Warb. ex K. Schum. & Lauterb.), bamboos (*Bambusa* spp.), and sugar cane (*Saccharum officinarum* L.). We recorded one version of the lunar calendar of Aneityum (Table 1), but there are likely others. This version was recorded in conversations with the late David Nasauman, who referred to it as the "sugar cane calendar", his son Wopa Nasauman (Fig 6), Nelly Dele, and Roneta Wamad, all in Anelcauhat.

## Architecture and cosmology

Vernacular architecture in Southeast Asia has been interpreted as representing both the social structure and cosmology of a people [36]. The Aneityum system of governance is reflected in the design of a traditional sleeping house, in which the two main upright posts represent the two Paramount Chiefs, who traditionally resided in the interior of the island (at Anijinwei for the southern chiefdom, and at Anecro for the northern chiefdom). Each post is forked at the top, representing the four districts. The cross beam supported by these posts, called *Naliañ atau atimi,* represents the tribe of the Holy Priests and simultaneously a physical *kastom* road that connects the two seats of the Paramount Chiefs (at Anijinwei and Anecro). Only the Holy Priests are permitted to walk this particular *kastom* road, and this beam represents the reconciliation between the two paramount chiefs, as facilitated by the Holy Priests. These architectural details reinforce the cultural importance of the paramount chiefdoms. Plant-based houses (Fig 6)—now being revived on Aneityum—connect closely with the ecological calendar which guides the seasonality and timing for collecting plants used in construction.

**Table 1. Anejoم̃ lunar calendar.**

| Gregorian month | Anejoم̃ month | Translation of name | Details |
|---|---|---|---|
| January | *Niviyeng* | "Big wild sugar cane" | This is the time of year with lots of wild cane, ready to make flowers soon. |
| February | *Inmohoc u nupudec* | "Month/moon of Nubidou (a village in the east)" | This is the time for planting gardens at Idec, which is the doorway area for the Paramount Chief at Anecro. |
| March | *Niau* | "Small bamboo cane" | This is when *niau* has flowers, which is when you plant all crops. |
| April | *Neplarou* | "In-between" | This is a transition time between good and bad weather seasons, from January to March and May and beyond. End of planting season. |
| May | *Netohas* | "Bad sugar cane" | If you plant sugar cane this month, the quality will be very poor. |
| June | *Netopni* | "Good sugar cane" | This is the time when sugar cane should be planted. |
| July | *Nowoyeng* | "Fruit of wild cane" | This is a good time to plant root crops. Expert David Nasauman gave a second meaning for Nowoyeng: *nowo* = a kind of tree, *yag* = yellow; thus, the time a certain tree, Gyrocarpus americanus (=nowo), has yellow (=yag) leaves which then fall. This is the time that the [large] clamshell has eggs. |
| August | *Niegacen, or Niag acen* | "Sour wild cane" | This is the time when the water inside wild cane (which is drunk when the weather is dry) is sour; it is a good time to plant taro. |
| September | *Imohocla* | "Month of clear weather" | During this month, the weather is very clear and sunny with few clouds, and a good time to harvest crops. |
| October | *Nisjiñjiñ* | "Time when things are confused" | If you plant things at this time, you don't know if the crops will turn out well or poorly. |
| November | *Nevatpou* | "Big sight" | When you look at plants in the bush, there is lots of wild food. You can relax, you don't have to work too hard. |
| December | *Nelvatpou* | "Small sight" | There is not much wild food in the bush. |

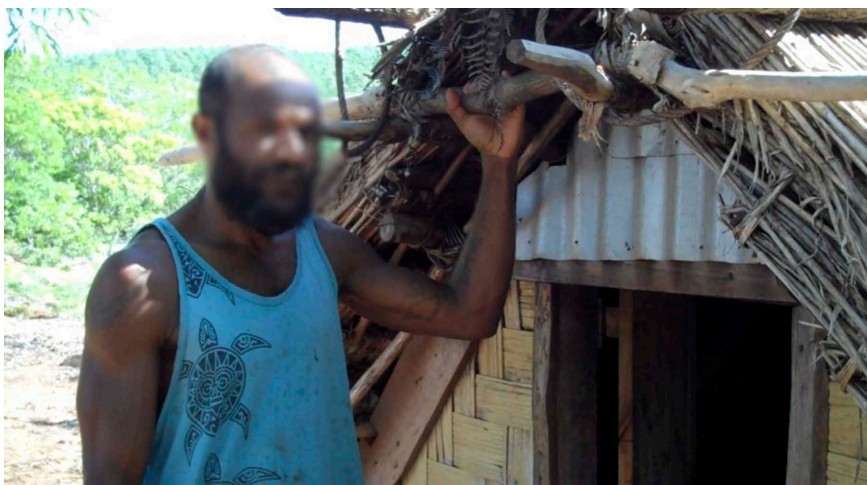

**Fig 6. Wopa Nasauman with his cyclone house built almost entirely of plants harvested at appropriate times according to the lunar calendar.** (video by K. David Harrison). See https://youtu.be/dUtNo0zdh0I?si=2ZiunZMj_z80nxYJ.

## Garden tabus

Proscriptions against taking certain actions before or during gardening work are common and closely followed on Ane-ityum. They usually involve the consumption (or avoidance) of particular foods. If a gardener eats a particular food (e.g., food cooked with coconut milk), he may not go into certain types of garden (e.g., a taro garden "swamp", for a period of

2–3 days). Similarly, one who eats *nowat* (surgeonfish; *Acanthurus triostegus* and *Acanthurus triostegus*) shall not enter the garden for the same period. These *tabus* reflect local concern with keeping the gardens spiritually "clean".

One garden *tabu* that is especially strong involves the preparation of *namarai* [14:95], a fermented emergency food for times of disaster. It is prepared by mashing banana (*Musa* spp.), of which the Aneityumese recognize over 44 varieties [14:118] or breadfruit [*Artocarpus altilis* (Parkinson) Fosberg], of which the Aneityumese distinguish more than 30 varieties [14:118], and sometimes manioc (*Manihot esculenta* Crantz). The mash is mixed with water and placed in an underground pit, enclosed within layers of *Heliconia indica* Lam. leaves to ferment. A pit properly constructed and maintained can safely preserve some foods for up to a decade of continuous use (see for example, the discussion of 40–50 year old breadfruit pits in Pohnpei, Micronesia [37]). *Namarai* is related to the Sun, as it helps people to endure periods of failed crops, which in turn were caused by the Sun being offended and destroying the gardens. Perhaps due to this association, there is a strict proscription against entering the garden for five days following the consumption of *namarai*. This product was a critically important famine food on Aneityum in times past, but today it is made much less frequently.

## Discussion

Through these stories and contemporary practices, we hope to convey the important role that the Sun and Moon play in Aneityumese *kastom*. Both celestial bodies are temporal guides that, through ecological and lunar calendars, as well as their movements in reference to local geography, ensure that the agricultural cycle is well followed throughout the year. The image of the Walking Sun represents not only the traditional governance system that is held to be the root of Aneityumese identity but also conveys metaphorically the division of the year into seasons of long and short days. Through the sacred geography of the island, the complex time-reckoning tradition, architecture, and garden tabus, we see that the biocultural practices of Aneityum are strongly intertwined with the Sun and Moon as interactive beings. Further, it is clear that efforts to eradicate the pre-Christian relationships with Aneityum's sacred landscape [13] and to transform the way that locals conceptualize time [17] were not fully successful. Rather, Gardner's [16] view that missionization on Aneityum was

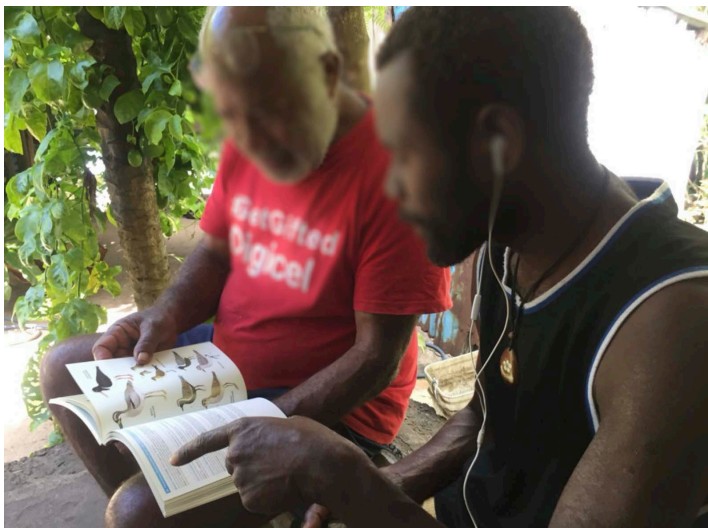

**Fig 7. Culture expert Jack Keitadi (left) discusses bird lore with Aneityum Island resident Chris Nevehev (right) in 2018.** (photo by K. David Harrison). See https://youtu.be/xMH6r8saefA?si=tIR3msj58nKDTzJu.

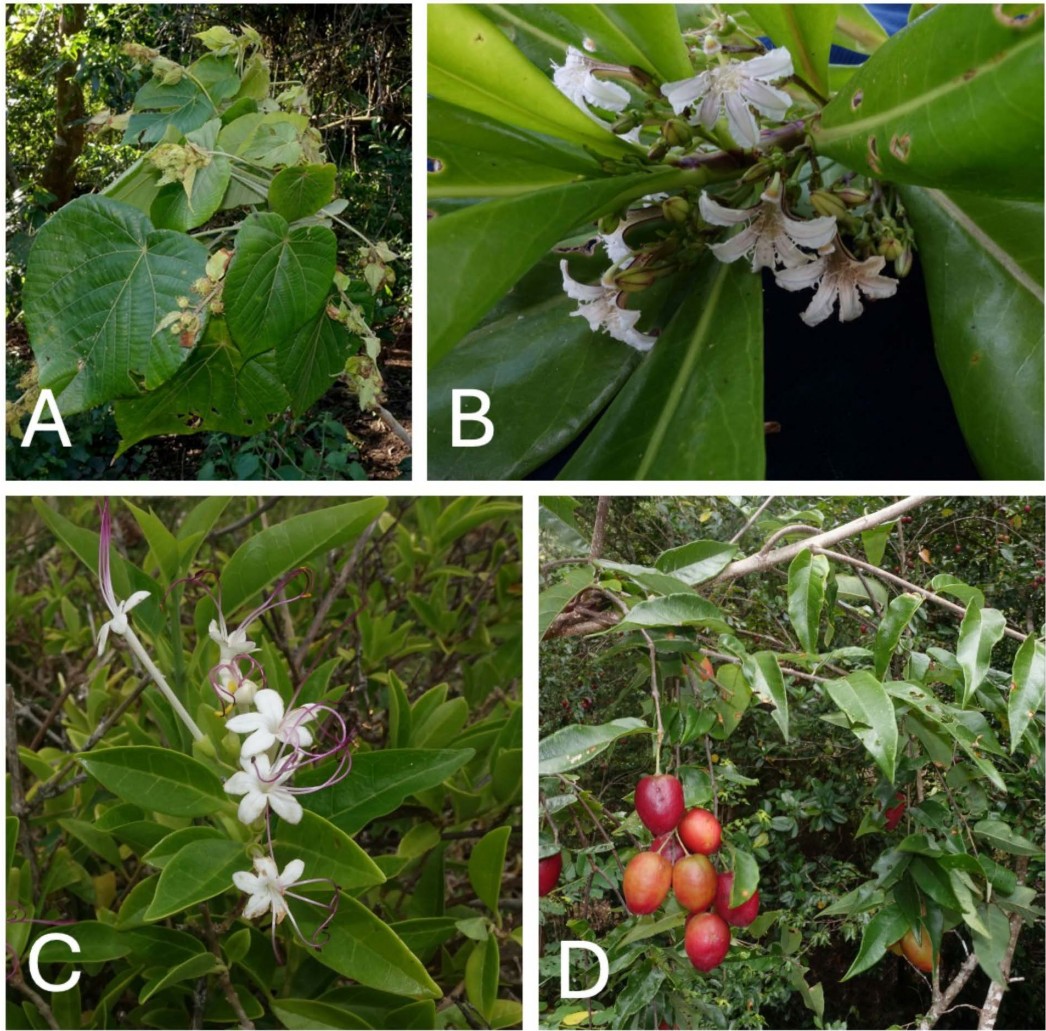

**Fig 8. A: *Macaranga dioica* (*GMP 3213*), B: *Scaevola taccada* (*GMP 3219*), C: *Volkameria inermis* (*GMP 3557*), D: *Aceratum oppositifolium* (*GMP 3635*).** Collections and photos by Gregory M. Plunkett (initials and numbers following each species name represent herbarium vouchers of each collection, which were deposited in the Vanuatu National Herbarium and the New York Botanical Garden Herbarium, among others).

one of translation and dialogue—one in which certain messages were either adopted or ignored—is likely to be a better model for what happened during this process.

The Sun and Moon are critical actors in stories of both the mythical past and the historical past—the subsequent waves of settlement on Aneityum shaped a culture that centers the Sun, in particular, as a powerful spirit. Islanders' stories of past Sun worship on Aneityum mark the island as unique within the religious and cosmological context of Melanesia. The previous literature on Aneityum, whether missionary or anthropological (e.g., [14,18]), does not mention Sun worship as a centralized or formalized religious practice. However, that literature clearly paints the Sun as an important *natmas* [spirit], and the pre-Christian practices of prayer and supplication to spirits [15] supports some level of practice resembling worship. Individuals from both Paramount Chiefdoms report historical Sun worship, and culture expert Ruben Neriam is steadfast in his claim that *Nagesega Rada* stood above the other *natmas* in a cosmological hierarchy.

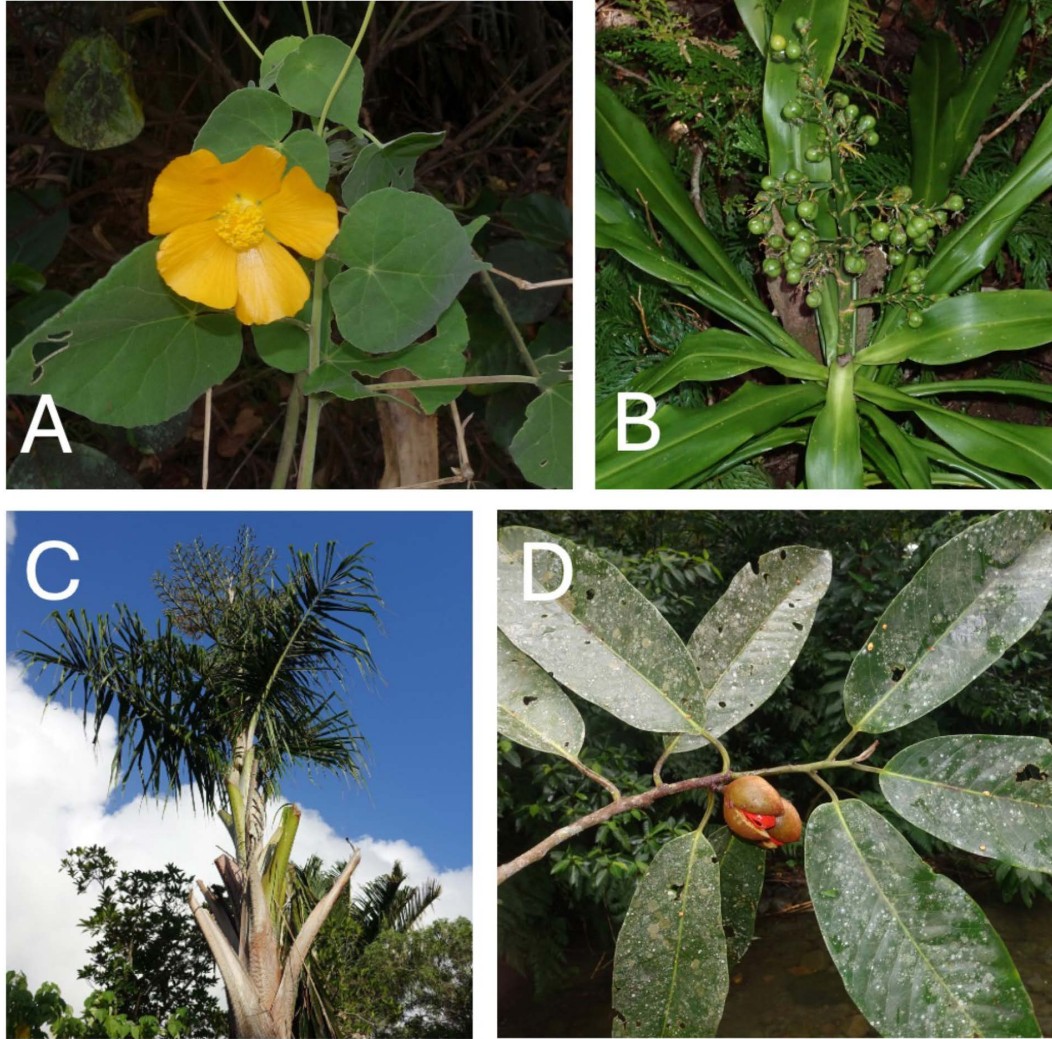

**Fig 9. A: *Abutilon indicum* (*GMP 3560*), B: *Dracaena* sp. nov.** (*GMP 3628*), C: *Metroxylon warburgii* (*GMP 3609*), D: *Myristica inutilis* (*GMP 4017*). Collections and photos by Gregory M. Plunkett (initials and numbers following each species name represent herbarium vouchers of each collection, which were deposited in the Vanuatu National Herbarium and the New York Botanical Garden Herbarium, among others).

Given the central importance of Aneityum's traditional governance structure and its gradual erosion over the last two centuries [18], it is unsurprising that many cultural leaders on Aneityum are working to revive it, along with associated cultural knowledge. *Kastom* revival is seen as an important strategy to increase local resilience, particularly important in the face of global change. Elders who see that the art of *namarai* production (Fig 5) is being lost are working to revitalize this practice, alongside efforts to replant the breadfruit trees that were once a very important food on the island. They are making a concerted effort to pass on environmental knowledge (Fig 7) (Fig 8) (Fig 9), including the details of canoe building, ecological calendars [35], foodways, navigation [18,38], plant-based medicine [28], traditional fishing methods, vernacular architecture, and weather magic [34]. These strategic cultural-preservation efforts have the practical effect of increasing resilience [39] in an era of an ever more unpredictable climate in the South Pacific, and with the respect for traditional practices (e.g., strict adherence to the limits of various activities to particular seasons), they also support sustainable living

on the island. In his 2018 interview with the authors, culture expert Jack Keitadi of Anelcauhat expressed optimism for the future:

> I think Aneityum's culture is resilient in that with a lot of changes coming in, people go (along) with it, but still retain their culture basically. Aneityum knows its culture…to respect the land you have to know your culture, that people have been surviving here for the last 3,000 years. If there's [climate] change, people will change the culture but still retain gardening methods, housing methods, and knowledge. (https://youtu.be/EJueu56-UK8?si).

## Supporting information

**S1 Appendix.  Special plant uses related to Sun and Moon lore.** Collection number refers to plant vouchers (herbarium specimens) used during interviews; initials and numbers refer to the collection numbers of the following collectors: AAM (Ashley A. McGuigan); GMP (Gregory M. Plunkett); MJB (Michael J. Balick). All plant specimens may be accessed in digital format at the CV Starr Virtual Herbarium, New York Botanical Garden, https://sweetgum.nybg.org/science/vh/. (DOCX)

## Acknowledgments

We gratefully acknowledge the people of Aneityum who have shared their knowledge with us. In addition to those named in the text, colleagues from Aneityum who assisted us in the present research include Kirk Keitadi, Kenneth Keith, Tony Keith, Titiya Lalep, Osiani Neriam, Chris Nevehev, Romario Yaufati, Natau Kenneth, Manapen Iauko, Tom Johnson.

## Author contributions

**Conceptualization:** K. David Harrison, Gregory M. Plunkett, Michael J. Balick.

**Data curation:** K. David Harrison, Neal Kelso, Dominik M. Ramík, Nadine Ramik, Gregory M. Plunkett, Wina Nasauman, Wopa Nasauman, Michael J. Balick.

**Formal analysis:** K. David Harrison, Michael J. Balick.

**Funding acquisition:** K. David Harrison, Michael J. Balick.

**Investigation:** Dominik M. Ramík, Nadine Ramik, Gregory M. Plunkett, Reuben Neriam, Wina Nasauman, Wopa Nasauman.

**Methodology:** K. David Harrison, Neal Kelso, Dominik M. Ramík, Gregory M. Plunkett.

**Writing – original draft:** K. David Harrison, Neal Kelso, Michael J. Balick.

**Writing – review & editing:** K. David Harrison.

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
